# Programmed Death-Ligand (PD-L1), Epidermal Growth Factor (EGF), Relaxin, and Matrix Metalloproteinase-3 (MMP3): Potential Biomarkers of Malignancy in Canine Mammary Neoplasia

**DOI:** 10.3390/ijms25021170

**Published:** 2024-01-18

**Authors:** Makchit Galadima, Mariana Teles, Josep Pastor, Javier Hernández-Losa, Joan Enric Rodríguez-Gil, Maria Montserrat Rivera del Alamo

**Affiliations:** 1Department of Animal Medicine and Surgery, Faculty of Veterinary Medicine, Universitat Autònoma de Barcelona, 08193 Bellaterra, Spain; makchitgaladima@gmail.com (M.G.); josep.pastor@uab.cat (J.P.); juanenrique.rodriguez@uab.cat (J.E.R.-G.); 2Department of Cell Biology, Physiology and Immunology, Universitat Autònoma de Barcelona, 08193 Barcelona, Spain; 3Department of Pathology, Hospital Universitari Vall d’Hebron, VHIR, 08035 Barcelona, Spain; javier.hernandez@vallhebron.cat

**Keywords:** canine mammary neoplasia, gene expression, malignancy biomarkers, programmed death-ligand 1, epidermal growth factor, relaxin, matrix metalloprotease-3

## Abstract

Gene expression has been suggested as a putative tool for prognosis and diagnosis in canine mammary neoplasia (CMNs). In the present study, 58 formalin-fixed paraffin-embedded (FFPE) paraffined canine mammary neoplasias from 27 different bitches were included. Thirty-seven tumours were classified as benign, whereas thirty-one were classified as different types of canine carcinoma. In addition, mammary samples from three healthy bitches were also included. The gene expression for vascular endothelial growth factor-α (*VEGFα*), *CD20*, progesterone receptor (*PGR*), hyaluronidase-1 (*HYAL-1*), programmed death-ligand 1 (*PD-L1*), epidermal growth factor (*EGF*), relaxin (*RLN2*), and matrix metalloproteinase-3 (*MMP3*) was assessed through RT-qPCR. All the assessed genes yielded a higher expression in neoplastic mammary tissue than in healthy tissue. All the evaluated genes were overexpressed in neoplastic mammary tissue, suggesting a role in the process of tumorigenesis. Moreover, PD-L1, EGF, relaxin, and MMP3 were significantly overexpressed in malignant CMNs compared to benign CMNs, suggesting they may be useful as malignancy biomarkers.

## 1. Introduction

Early diagnosis of tumours, especially those with malignant characteristics, is one of the main milestones in current cancer therapy strategies. Thus, early diagnosis allows the implementation of a precocious treatment and, consequently, the improvement of the prognosis for the patient. Nowadays, the study of tumour-linked gene expression patterns has gained relevance as a useful tool in the diagnosis and prognosis, and especially the treatment approach, of canine mammary neoplasia (CMNs) [1]. Several of these biomarkers have been widely evaluated in breast cancer, but research specifically focused on CMNs is less abundant.

In human breast cancer (HBC), gene expression has allowed the categorization of them into different molecular subtypes [2], consequently providing a more accurate prognosis and therapy. Although a few steps behind, veterinary medicine has also assessed the expression of diverse potential biomarkers for CMNs. Thus, the expression of both progesterone and oestrogen receptors was evaluated through immunohistochemistry (IHC) techniques in the late 1990s [3]. With the addition of the epidermal growth factor to the equation, a molecular typing for CMNs was initially proposed later on [4]. Through the years, different techniques, such as microarrays or gene sequencing, have been developed to improve the diagnosis, the prognosis, and the therapeutic approach in CMNs.

RNA sequencing has provided a deep knowledge of the molecular biology of cancer [5,6,7]. RT-qPCR is considered an important tool for transcriptomic analysis in a variety of studies, including those carried out with samples from *Canis lupus familiaris* mammary tumours [8,9]. In the present study, the stability of four potential reference genes was evaluated to be used in the normalization of the target genes. According to previous studies, the following reference genes were used: glyceraldehyde 3-phosphate dehydrogenase (*GAPDH*), hypoxanthine-guanine phosphoribosyltransferase (*HPRT*), ribosomal protein S19 (*RPS19*), and ribosomal protein L8 (*RPL8*) [10,11,12,13].

This study aimed to assess the gene expression in CMNs of hyaluronidase-1 (*HYAL-1*), *CD20*, matrix metallopeptidase 3 (*MMP3*), vascular endothelial growth factor-α (*VEGFα*), relaxin 2 (*RLN2*), programmed death-ligand 1 (*PD-L1*), epidermal growth factor (*EGF*), and progesterone receptor (*PGR*) to determine their putative potential as molecular biomarkers of malignancy in CMNs through qRT-PCR.

## 2. Results

### 2.1. Tumours Typification

The histological typification of the 58 tumours included in the study yielded a total of 37 benign CMNs, with 21 being classified into different categories of malignant tumours (Table 1). Likewise, healthy samples were also confirmed by histological evaluation.

### 2.2. Estimation of Gene Efficiency

The efficiency of each gene was estimated by a standard curve generated by using serial dilutions of a pool of cDNA samples. The correlation coefficients were highly linear, and the PCR amplification efficiencies were close to 100% for all the primers (Table 2). From the four reference genes evaluated to be used in the normalization of data, the *HPRT* gene was eliminated from the evaluation of best reference genes since, according to CFX Maestro™ Software data, HPRT expression was unstable and therefore it cannot be used as a reference gene. A ranking according to the expression stability of each gene in a given sample set and experimental design is performed with both algorithms. According to NormFinder, stability values for candidate reference genes were 0.064 for *GADPH*, 0.041 for *RPL8*, and 0.028 for *RPS19*. The latter was then considered the most stable reference gene according to the NormFinder algorithm. According to the reference gene selection tool (CFX Maestro™ Software, BioRad Version 5.3.022.1030, year 2021 Hercules, CA, USA), the stability values for candidate reference genes were 1.07 for GADPH, 0.57 for *RPL8*, and for *RPS19*. Therefore, the combination *RPL8* + *RPS19* was used for RT data normalization.

Concerning the expression levels of the target genes evaluated in the present study, *HYAL1*, *VEGFα*, *CD20*, and *PGR* genes presented a similar pattern of response, showing significantly higher levels of expression both in benign and malignant tumours when compared to the control group. However, no statistically significant difference in the expression levels was observed when comparing both types of tumours, benign and malignant. Concerning the mRNA levels of *PDL-1*, *EGF*, *RLN2*, and *MMP3*, a similar pattern of response was also depicted. Expression in both benign and malignant tumours was significantly up-regulated when compared to the control group. Moreover, malignant tumours presented significantly higher expression levels when compared to benign tumours (Figure 1).

### 2.3. Correlation between Genes

Several correlations between genes were observed. Table 3 and Table 4 show the correlation between genes in benign and malignant CMNs, respectively. In this sense, gene expression for PGR was positively correlated with *CD20*, *EGF*, *RLN*, and *MMP3* gene expression in both benign and malignant CMNs. Gene expression for EGF was positively correlated to *RLN2* and *MMP3* in both benign and malignant CMNs. Finally, gene expression for *RLN2* was positively correlated with *CD20* and *MMP3* in both benign and malignant CMNs. Focusing on benign CMNs, gene expression for *VEGFα* was negatively correlated with relaxin and *MMP3*. Gene expression for *EGF* was positively correlated to *CD20* and HYAL1 expression. Regarding malignant CMNs, gene expression for CD20 was positively correlated to *MMP3* gene expression. Finally, gene expression for *PD-L1* was negatively correlated with *HYAL1* and *EGF* expression.

Following the evaluation of the linear correlation, a principal components analysis (PCA) was performed. Table 5 shows the relative weighting for the first three components. The first principal component accounted for 80.9% to 88.4% of the total variance. The contributing molecules were *CD20*, *PGR*, *EGF*, *RLN2* and *MMP3*. In the second component, two molecules overweighted, namely *HYAL1* and *PD-L1* with 71.6% and −81.2%, respectively. Finally, in the third component, only *VEGFα* was over-represented, being responsible for 86.3% of the total variance. Figure 2 shows the spatial expression of the different components analysed.

## 3. Discussion

The present results highlight the differential gene expression of canine mammary neoplasia compared to healthy mammary tissue. The genes evaluated in this study may be classified into two different categories: those related to the presence of mammary neoplasia (namely VEGFα, CD20, PGR, and HYAL1) and those indicative of malignancy (namely PDL-1, EGF, RLN2, and MMP3).

Neovascularization is mandatory in tumorigenesis, tumour growth, and dissemination [14]. Neovascularization is regulated, among other angiogenic factors, by the vascular endothelial growth factor (VEGF) [15]. Thus, the role of VEGF in tumorigenesis is more than expected. VEGF expression has been widely demonstrated to be directly correlated with tumour growth in breast cancer [16,17,18,19,20]. Focusing on the canine mammary neoplasia, the expression of VEGF has been assessed in both serum and tissue samples. VEGF is over-expressed in bitches with malignant CMNs in comparison with healthy bitches [21,22,23,24]. According to the literature, malignant CMNs also over-express VEGFα when compared with benign CMNs [21]. However, the present study shows no statistical difference in the expression of VEGFα between benign and malignant CMNs. A feasible explanation for this difference is the different applied techniques. Thus, in the present study, VEGFα expression was assessed through RT-qPCR, whereas in the previous study, VEGFα expression was evaluated using immunohistochemistry. The higher expression of VEGFα in both benign and malignant CMNs suggests that this molecule is involved in the neovascularization processes of CMNs regardless of their benign or malignant nature.

CD20 has been scarcely described in CMNs. To the best of the authors’ knowledge, CD20 expression has been described in only one case of canine primary mammary lymphoma [25], with no other research performed in this field. Peripheral levels of CD20 have been previously assessed and compared among healthy bitches, bitches carrying a malignant CMN, and bitches carrying a benign CMN by this research group [26]. In that previous study, higher levels of serum CD20 were observed in bitches with CMNs regardless the neoplasia was benign or malignant, with no significant difference among them. These results agree with the present study, where tissue over-expression of CD20 was observed in CMNs in comparison with healthy tissue, with no significant difference between benign and malignant neoplasia. CD20 has been studied in human breast cancer, but its role is not clear. Whereas some studies suggest that CD20 is associated with a good prognosis in human breast cancer [27], others associate it with a bad prognosis [28,29]. In the present study, no such affirmation can be made since most of the patients included did not provide any further feedback after the surgery. Focusing on the present results, it seems clear that CD20 plays a role in canine mammary tumorigenesis, but it is not helpful as a diagnostic biomarker.

The role of ovarian hormones on CMNs, oestrogen and progesterone, as well as their respective receptors, is well-known and described in the literature [30]. Regarding progesterone receptor (PGR), the literature describes its presence in mammary tissue as a favourable prognostic indicator [31,32]. The expression of PGR has been assessed in healthy and neoplastic canine mammary tissue through different techniques, always yielding similar results. According to the literature, PGR is expressed by all benign CMNs and approximately two-thirds of malignant MCTs [31,33,34]. However, the present results disagree with those previously reported. In this study, the overall expression of PGR did not differ between benign and malignant CMNs. A feasible explanation is the fact that, in the present study, malignant CMNs were not classified according to the degree of malignancy due to the low number of samples. According to the literature, simple carcinomas show a higher expression of PGR in comparison with complex carcinomas [35], the lack of differentiation between simple and complex carcinomas in our study would dilute these possible differences.

Hyaluronidases are enzymes which degrade hyaluronic acid, the major constituent of the extracellular matrix (ECM) [36]. The degradation of ECM is a requirement for tumour development, so a putative role of hyaluronidases in tumour development is expected. The gene expression of the hyaluronidase isoform HYAL-1 has been scarcely described in CMNs and the results are controversial. Thus, whereas Varallo et al. [37] have described a higher expression of HYAL-1 in canine mammary carcinoma when compared with healthy tissue, Sakalauskaite et al. [9] observed no difference between carcinoma and healthy adjacent tissue, except in German Sheperd bitches, which showed a higher expression of HYAL-1 in the adjacent tissue. According to the authors’ knowledge, this is the first study to compare the expression of HYAL-1 among healthy mammary tissue, and benign and malignant mammary tumours in bitches. The present results partially agree with those previously obtained by Varallo et al. [37] since malignant CMNs showed a higher expression of HYAL-1 when compared with healthy mammary tissue. However, no significant difference was observed between benign and malignant CMNs. The present results clearly suggest that HYAL-1 may be involved in mammary tumorigenesis, but it is not useful as a malignancy biomarker, being its action probably related to the modification of the ECM. HYAL-1 degrades hyaluronic acid, generating small fragments of hyaluronic acid which are angiogenic [38,39]. In vitro studies have also demonstrated the contribution of HYAL-1 in tumour cell proliferation, motility, invasion, tumour growth, metastasis, and angiogenesis in breast, bladder, prostate, and colon cancer [40,41,42,43,44].

PD-L1 (programmed death-ligand 1) is a transmembrane protein which is expressed on the surface of activated cytotoxic T cells. This molecule inhibits the production of IL-2 and the migration and proliferation of T cells [45]. It also activates the PD-1 protein localized on the surface of T cells, which would lead to immune tolerance [46]. When engaged, the PD-1/PD-L1 complex induces the dysfunction, exhaustion, and neutralization of TILs [47]. TILs are known to be recruited to the tumour microenvironment to kill tumoural cells. Consequently, the overexpression of PD-L1 is considered a protective mechanism developed by neoplasia [48]. The studies assessing its possible role in mammary cancer in women have yielded discordant results. Thus, some studies have observed that this molecule is commonly expressed in triple-negative breast cancers (TNBC) [49] and its higher expression is related to a significantly decreased survival and higher tumour grade [50], whereas a more recent study has shown that only 10% of TNBC are positive to PD-L1 [51].

Focusing on CMNs, the expression of PD-L1 has been scarcely studied. It has been described to be expressed in approximately two-thirds of both benign and malignant CMNs through immunohistochemistry techniques [52]. Likewise, the gene overexpression of PD-L1 has been correlated to metastatic malignant CMNs and shorter survival periods [53]. According to the authors’ knowledge, this is the first time that gene expression of PD-L1 has been compared between benign and malignant CMNs. In the present study, PDL-1 was significantly over-expressed in CMNs compared to healthy mammary tissue, with malignant CMNs showing the highest tissue expression. These results are suggestive of the putative potential of PD-L1 as a malignancy biomarker of CMNs.

Epidermal growth factor (EGF) is a protein which is involved in the proliferation and differentiation of epithelial cells [54] in addition to angiogenesis, metastasis, and apoptosis inhibition [55,56]. The binding of EGF with its receptor (EGFR) induces changes in gene transcription that enhance tumorigenesis [57]. Likewise, EGF overexpression has been associated with tumour invasion and progression [58,59,60]. Focusing on CMNs, in vitro studies have demonstrated that EGF enhances proliferation, chemotaxis and the production of VEGF, and decreases apoptosis in canine carcinomas [61]. However, in vivo studies on EGF have yielded controversial results. Thus, whereas Klopfleish et al. [62] observed a down-regulation in the gene expression of EGF in metastatic canine mammary carcinomas, Queiroga et al. [63] reported higher concentrations of EGF in neoplastic mammary tissues by enzyme-immunoassay determinations. In the present study, CMNS showed significantly higher expression of EGF than healthy mammary tissue. As stated above, EGF is involved in tumorigenesis. Thus, overexpression was somehow expected in neoplastic tissue. On the other hand, malignant CMNs showed significantly higher expression than benign CMNs, making EGF a potential biomarker for malignancy.

Relaxin is a peptide hormone that, among other functions, promotes both the growth and development of the mammary gland, which are mandatory events to ensure lactation after delivery [64,65]. A not-so-well-known role of this hormone is the one played in mammary neoplasia. In this regard, relaxin enhances the invasiveness of breast cancer in vitro [66] and higher serum levels of relaxin have been related to poor prognosis of breast cancer in women [67]. Both serum and tissue expressions have been assessed also in CMNs. However, the results are contradictory. Thus, while some studies [68,69] have observed no correlation between relaxin expression and tumour malignancy, some others have found that relaxin is over-expressed in malignant III-grade CMNs [10]. In the present study, CMNs showed significantly higher expression of RLN2 in comparison with healthy canine mammary tissue. Additionally, malignant CMNs significantly overexpressed RLN2 when compared to benign CMNs. These results demonstrate the potential of RLN2 as a malignancy biomarker in CMNs.

MMPs are proteases that play a crucial role in many biological processes such as the remodelling process of the extracellular matrix, cell proliferation, migration, and differentiation, and tissue invasion and vascularization (see [70] for a review). These biological processes must be accurately balanced. Otherwise, they may cause pathological conditions, including cancer and tumour progression (see [70] for a review). In fact, MMP members are known to play a relevant role in cancer progression since they facilitate the invasion and metastasis of the original tumour by dissolving the basement membrane and degrading the extracellular matrix [66].

MMPs have been described to be present in the mammary tissue and linked to the extensive remodelling that the mammary gland undergoes from the pre-puberal to the adult stage, and during pregnancy and lactation (see [71] for a review). MMP3, also known as stromelysin-1, has been described to play a relevant role in the development of the mammary gland, specifically in the lateral branching of ducts during the phases of mid-puberty and early pregnancy [72]. This enzyme has been reported to be involved in angiogenesis, cell growth and cell invasion [73]. These specific features have warranted further research on MMP3′s role in tumourigenesis; however, this topic is beyond the aims of the present manuscript. Regarding breast cancer, MMP3 has been described to be over-expressed in neoplastic mammary tissue [73], to be involved in both mammary cancer invasion and metastasis [74,75,76,77], and has also been considered a prognostic factor [78,79].

Focusing on CMN, according to the consulted literature, over-expression of MMP3 has been described in serum [80] and tissue [69] samples from bitches with mammary carcinoma. The present results confirm that MMP3 plays a role in CMN development since it is over-expressed in both benign and malignant mammary tumours. On the other hand, the significantly higher expression in malignant CMNs suggests that MMP3 may be involved in further metastatic processes and could be used as a prognostic parameter, agreeing with those findings observed in women. Unfortunately, it was not possible to acquire the follow-up data for the patients included in this study, making it impossible to establish a correlation between survival and MMP3 expression. Thus, further research on this molecule is warranted.

Statistical correlations among the different evaluated gene expressions were also assessed. Molecules such as PGR, CD20, EGF, relaxin, and MMP3 showed significant correlations between them in both benign and malignant CMNs. These correlations suggest that they might be involved in the process of mammary tumorigenesis, but not necessarily in the process of malignity. Focusing on correlations in benign CMNs, it is worth mentioning the negative correlation of VEGFα with relaxin and MMP3. As indicated above, neovascularization is needed for tumorigenesis, and tumour growth and dissemination [14]. This is especially relevant to malignant tumours, which greatly depend on the neovascularization to metastasize [81]. Thus, a negative correlation is not surprising in benign CMNs.

Regarding malignant CMNs, a significant negative correlation of PD-L1 with HYAL1 and EGF was also observed. Since PD-L1 has been described to act as a protective mechanism developed by neoplasia [48], this negative correlation is surprising and no feasible explanation can be provided, warranting more research.

Focusing on the PCA, the relevant elements of component 1 are the same molecules which already showed a significant correlation in the previous linear analysis, namely CD20, PGR, EGF, relaxin and MMP3. These results suggest that the combined analysis of these markers, further highlighted by the PCA analysis, may have differential diagnostic potential in CMNs between normal and neoplastic tissue.

The assessment of gene expression in CMNs is also interesting for personalized therapeutic approaches. Some of the evaluated molecules have been suggested or are already being used as therapeutic targets for human cancer. One of these potential targets is PD-L1. Previous studies in human cancer have shown that the blockade of PD-L1 with monoclonal antibodies enhances the destruction of cytotoxic T-lymphocyte-mediated tumour cells and improves the activation of antigen presentation and cytokine release, suggesting a putative potential as a therapeutic target [82,83]. Similar results have been observed in in vitro studies in canine B cell lymphoma [8,84] and oral melanoma [85]. Since PD-L1 is significantly over-expressed in malignant CMNs, it seems logical to hypothesize that the blockade of PD-L1 may be a potential therapeutic target for CMNs.

EGF has been a targeted molecule in breast cancer in women since 1998 (HER2), when the first anti-HER2 antibody, trastuzumab, was approved [86]. Several drugs have been further developed since then and are commercially available to treat human mammary cancer (see [87] for a review). The implementation of anti-HER2 antibodies as targeted therapies has improved the outcome of breast cancer patients [87]. Therefore, the implementation of these drugs in CMNs may also be beneficial.

VEGFα and MMP3 are also promising molecules as therapeutic targets. These molecules are involved in neovascularization, angiogenesis, cell growth and invasion [15,73]. These biological processes are fundamental for tumour growth and dissemination, thus their blockade may prevent the expansion of the tumour.

Another molecule that warrants further research is PGR. Recent studies have demonstrated that the administration of aglepristone, a progesterone receptor antagonist, inhibits the proliferation index in PGR-positive canine carcinomas [88] and increases the disease-free period [89].

Relaxin has been tested as a co-adjuvant drug in the treatment of cancer, but not as a targeted molecule itself for inhibiting tumour growth. This protein increases the expression and catalytic activity of some MMPs [90,91,92,93], which ultimately would facilitate the degradation of the tumour stroma and, consequently, increase the porosity of the tumour to oncolytic drugs [94,95]. In this sense, a previous study has demonstrated that the intratumoral transplantation of tumour cells containing the relaxin gene or haematopoietic cells containing relaxin has an antitumour effect in tumour-bearing mice [96].

Finally, HYAL-1 has been shown to be involved in several biological processes related to tumorigenesis and metastasis such as cell growth, migration, invasion and angiogenesis in breast cancer [97,98]. Thus, blocking this molecule could hinder these processes and, consequently, be used as a treatment to target CMNs.

## 4. Materials and Methods

### 4.1. Animals and Sampling

In the present study, a total of 30 bitches referred to the Teaching Veterinary Hospital (Fundació Hospital Clínic Veterinari) at Universitat Autònoma de Barcelona (UAB, Catalonia, Spain) and diagnosed with mammary tumours were included. Patients were aged between 6 and 13 years (mean age: 10.3 years) and belonged to different dog breeds. All females were subjected to surgery to remove the mammary tumours. Once removed, the masses were fixed with 10% paraformaldehyde (Sigma-Aldrich, Barcelona, Spain) and submitted to the laboratory for tumour typification through hematoxylin/eosin histologic evaluation. Samples were further embedded in paraffin and four to five 10 μm sections from each paraffined tumour were obtained for RNA extraction.

A total of 58 mammary samples were submitted for histology purposes. In addition, healthy mammary tissue from 3 bitches was also included as a control group. Specimens were obtained according to the guidelines of the Ethical Committee of Animal Care and Research of the UAB (protocol CEEAH number 1127, 20 March 2012).

### 4.2. Extraction of RNA and Synthesis of Complementary DNA

Only paraffin-conserved samples with more than 80% of tumour samples confirmed by histology were used. After deparaffinization of 10 µm paraffin-embedded tissue samples, RNA was extracted using the High Pure FFPET RNA Isolation Kit (Roche, Basel, Switzerland). Cells were lysed in 100 µL RNA Tissue Lysis Buffer, 16 µL 10% SDS, and 40 µL of Proteinase K for 30 min at 85 °C and shaking at 600 rpm. After this step, 60 µL of Proteinase K was added and the tissue samples were incubated at 55 °C and shaken at 600 rpm for 30 min. After cell lysis, the samples were incubated for 15 min at 15 to 25 °C with the DNAase working solution to enhance the RNA extraction. After this, the samples were centrifuged several times to eliminate debris and, finally, 50 µL of RNA elution buffer was added to obtain the pure total RNA. RNA quantification (ng/µL) was conducted using a NanoDrop Spectrophotometer (Thermo Fisher Scientific, Waltham, MA, USA). Reverse transcription was performed using 1 μg of the total RNA using the iScript™ cDNA synthesis kit (Bio-Rad Laboratories, Hercules, CA, USA) according to the manufacturer’s instructions. The iScript cDNA synthesis kit is a sensitive and easy-to-use first-strand cDNA synthesis kit for gene expression analysis using real-time qPCR.

### 4.3. Real-Time Quantitative Polymerase Chain Reaction (RT-qPCR)

The set of genes studied includes indicators of malignancy (HYAL1, MMP3 and RLN2) and indicators of the hormonal response (PGR). Primers’ information is given in Table A1. The efficiency of amplification was tested for each primer pair as follows: 5-fold serial dilutions of the cDNA pool were analysed, and E = 10 (−1/s) was used as the formula for efficiency, where s is the slope generated by the serial dilutions. RT-qPCR was performed in a Bio-Rad CFX384 real-time PCR detection system (Bio-Rad Laboratories). Reactions were performed using iTaqTM Universal SYBR^®^ Green Supermix (Bio-Rad Laboratories) according to the manufacturer’s instructions. Briefly, 1 cycle at 95 °C for 5 min, 40 cycles at 95 °C for 30 s, 60 °C (or 55 °C for HYAL1 and VEGFA) for 30 s, and 72 °C for 30 s were run. Expression data, obtained from two independent biological replicates, were used to calculate the threshold cycle (Ct) value.

### 4.4. Standardization Strategy

NormFinder and the algorithm that comes with the CFX Maestro™ Software (Reference Gene Selection Tool, Bio-Rad Laboratories) were used to identify the most appropriate reference gene or combination of reference genes among four: GAPDH, HPRT, RPL8, and RPS19. Since the CFX Maestro™ Software was used for the calculations, we used the selection of the best combination references genes defined by the Bio-Rad software. Therefore, the expression levels of the target genes were normalized using the best combination of two reference genes and relative gene expression calculated with the ΔΔCt method using the CFX Maestro™ Software.

### 4.5. Statistical Analysis

The Shapiro–Wilk test was used to study the normality distribution between variables because some or all of the variables did not follow a normal distribution (*p* < 0.05). A non-parametric Kruskal-Wallis test with Dunn’s multiple comparison test was used to study if there were statistically significant differences between groups. The Spearman coefficient of correlation was studied between genes. Statistically significant differences were when *p* values were <0.05. All statistical analysis was performed using GraphPad Prism version 8.0.0 for Windows (GraphPad Software, San Diego, CA, USA, www.graphpad.com). IBM SPSS statistics version 22 (IBM, Tulsa, OK, USA) was used to study principal component analysis.

## 5. Conclusions

According to the present results, it can be concluded that the over-expression of the evaluated genes is related to the tumorigenic process in canine mammary neoplasia. Some of these genes, namely *PDL-1*, *EGF*, *RLN2*, and *MMP3*, may be useful as malignancy biomarkers in CMNs. Finally, the present results indicate that no combination of parameters provides a 100% specificity to differentiate between malignant and benign CMNs. Thus, the combination of more than one parameter is needed for that purpose. 

## Figures and Tables

**Figure 1 ijms-25-01170-f001:**
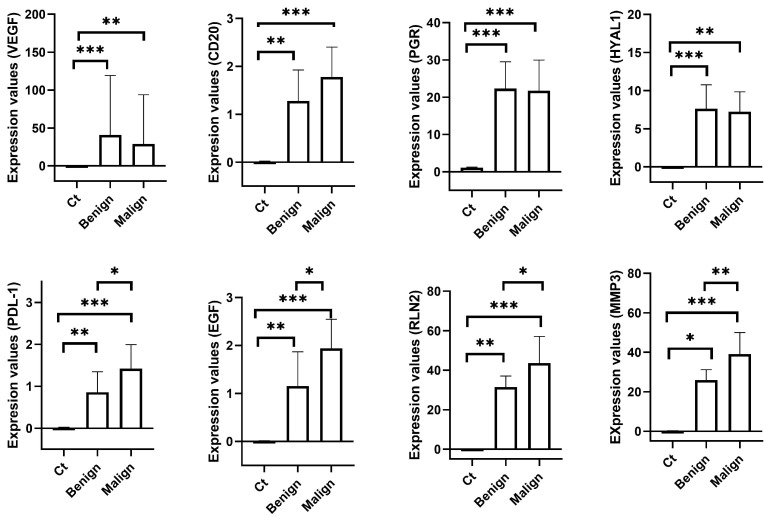
Relative mRNA levels of target genes measured in samples obtained from healthy mammary tissue and mammary tumours. Bars represent the median value with interquartile range. Statistical significances between groups are marked with asterisks (* *p* < 0.05; ** *p* < 0.01; *** *p* < 0.001). VEGF: Vascular endothelial growth factor α; PGR: Progesterone receptor; HYAL1: Hyaluronidase 1; PD-L1: Programmed death-ligand 1; EGF: Epidermal growth factor; RLN2: Relaxin 2; MMP3: Matrix metallopeptidase 3.

**Figure 2 ijms-25-01170-f002:**
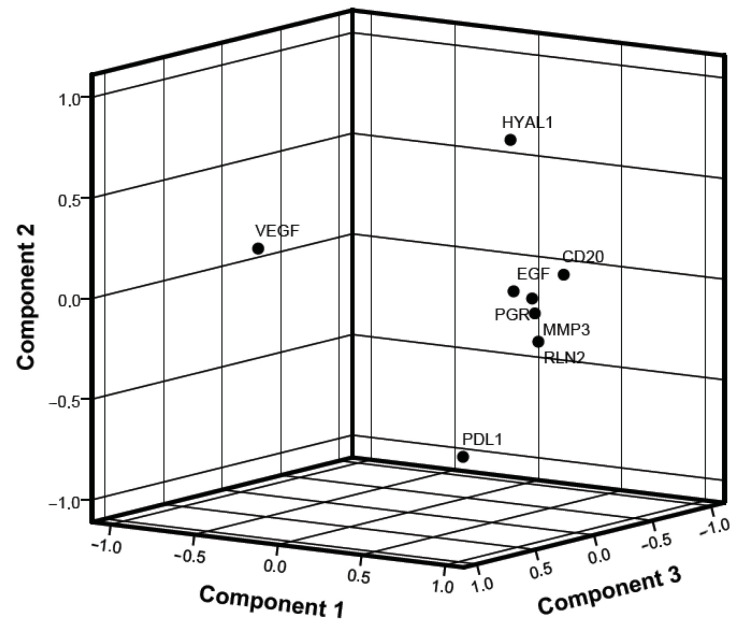
Spatial expression of the different components analysed.

**Table 1 ijms-25-01170-t001:** Histological classification of the mammary tumours included in the present study.

Malignant (n = 21)		Benign (n = 38)	
Typification	n	Typification	n
Complex adenocarcinoma	2	Adenoma	36
Complex carcinoma	8	Hemangioma	1
Solid carcinoma	1		
Anaplastic carcinoma	2		
Adenosquamous carcinoma	1		
Tubullo-papillary carcinoma	6		
Intraductal-papillary carcinoma	1		

**Table 2 ijms-25-01170-t002:** Amplification efficiency of primers in samples from *Canis lupus familiaris*. E*: Efficiency. The amplification efficiency of each primer was calculated according to the equation E = 10^(−1/slope)^.

Gene	Slope	R^2^	E*	E (%)
*GADPH*	−3.25	0.99	2.03	103
*HPRT*	−3.11	0.99	2.05	105
*RPL8*	−3.27	0.94	2.02	102
*RPS19*	−3.25	0.99	2.03	103
*HYAL1*	−3.16	0.99	2.07	107
*MMP3*	−3.21	0.99	2.05	105
*RLN2*	−3.29	0.99	2.01	101
*VEGFA*	−3.20	0.94	2.05	105
*PGR*	−3.25	0.99	2.03	103
*CD20*	−3.29	0.98	2.02	102
*PD-L1*	−2.87	0.98	2.23	123
*EGF*	−3.05	0.96	2.13	113

*GADPH*: Glyceraldehyde-3-phosphate dehydrogenase; *HPRT*: Hypoxanthine guanine phosphoribosyltransferase; *RPL8*: Ribosomal protein L8; *RPS19*: Ribosomal protein S19; *HYAL1*: Hyaluronidase 1; *MMP3*: Matrix metallopeptidase 3; *RLN2*: Relaxin 2; *VEGFα*: Vascular endothelial growth factor α; *PGR*: Progesterone receptor; PD-*L1*: programmed death-ligand 1; *EGF*: epidermal growth factor.

**Table 3 ijms-25-01170-t003:** Coefficients of linear correlation between gene expression in benign CMNs.

	*VEGF-α*	*CD20*	*PGR*	*HYAL1*	*PD-L1*	*EGF*	*RLN2*	*MMP3*
*VEGFα*	….							
*CD20*	-	….						
*PGR*	-	0.558 **	….					
*HYAL1*	-	-	-	….				
*PD-L1*	-	-	-	-	….			
*EGF*	-	0.790 ***	0.502 **	0.418 *	-	….		
*RLN2*	−0.467 **	0.546 **	0.435 *	-	-	0.718 ***	….	
*MMP3*	−0.546 **	-	0.464 *	-	-	0.500 **	0.507 **	….

Statistically significant correlations between gene expression are marked with asterisks (* *p* < 0.05; ** *p* < 0.01; *** *p* < 0.001). *VEGFα*: Vascular endothelial growth factor α; *PGR*: Progesterone receptor; *HYAL1*: Hyaluronidase 1; *PD-L1*: Programmed death-ligand 1; *EGF*: Epidermal growth factor; *RLN2*: Relaxin 2; *MMP3*: Matrix metallopeptidase 3.

**Table 4 ijms-25-01170-t004:** Coefficients of linear correlation between gene expression in malignant CMNs.

	*VEGF-α*	*CD20*	*PGR*	*HYAL1*	*PD-L1*	*EGF*	*RLN2*	*MMP3*
*VEGFα*	….							
*CD20*	-	….						
*PGR*	-	0.850 ***	….					
*HYAL1*	-	-	-	….				
*PD-L1*	-	-	-	−0.725 **	….			
*EGF*	-	-	0.702 ***	-	−0.566 *	….		
*RLN2*	-	0.725 **	0.754 ***	-	-	0.547 *	….	
*MMP3*	-	0.654 **	0.718 **	-	-	0.662 *	0.718 **	….

Statistically significant correlations between gene expression are marked with asterisks (* *p* < 0.05; ** *p* < 0.01; *** *p* < 0.001). *VEGFα*: Vascular endothelial growth factor α; *PGR*: Progesterone receptor; *HYAL1*: Hyaluronidase 1; *PD-L1*: Programmed death-ligand 1; *EGF*: Epidermal growth factor; *RLN2*: Relaxin 2; *MMP3*: Matrix metallopeptidase 3.

**Table 5 ijms-25-01170-t005:** Principal components matrix of gene expression in CMNs.

	Component		
	1	2	3
*VEGFα*	−0.292	0.295	0.863
*CD20*	0.838	0.136	−0.132
*PGR*	0.809	0.047	0.096
*HYAL1*	0.334	0.716	−0.397
*PD-L1*	0.296	−0.812	−0.046
*EGF*	0.825	0.111	0.278
*RLN2*	0.884	−0.152	0.151
*MMP3*	−0.292	0.295	0.863

## Data Availability

Data are available under request to the authors.

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
