# Peer review of "Programmed Death-Ligand (PD-L1), Epidermal Growth Factor (EGF), Relaxin, and Matrix Metalloproteinase-3 (MMP3): Potential Biomarkers of Malignancy in Canine Mammary Neoplasia"

_ijms, 2024, doi:10.3390/ijms25021170_

Round 1

Reviewer 1 Report

Comments and Suggestions for Authors

This is a nice and well-written study on canine mammary neoplasm gene expression. Some suggestions to improve it are presented below.

Line 15 – the word “Tumors” refers to increased volume despite often being related as neoplasia, technically it can be another pathology process (abscess, cysts, granuloma etc..). In the other hand, “cancer” refers to a malignant neoplasia. In this way, I believe the authors should review the sentence “Canine mammary tumors (CMTs) are the most frequent cancers in intact bitches”

Line 38 – Please modify canine mammary tumors by canine mammary neoplasia.

Lines 50-66 – This paragraph seems to be a mix of discussion about the importance of adopting adequate PCR standards and techniques plus some material and methods when describing the genes used to evaluate normalization. I´m particularly not used to finding this information in the intro.

Line 462 – Shapiro-Wilk results were assumed as non-parametric if the p-value was < 0.05 or below 0.005?

Line 76 – Despite the material and methods section giving an adequate amount of information regarding molecular biology techniques applied, I believe more detail should be provided in the MMs to inform how the authors assumed the histological classification shown in the table 1. Was any immunohistochemistry applied or the diagnosis was assumed only on HE histologic evaluation?

Figure 1 – if the data regarding gene expression was assumed as non-parametric, the right way to express them in graphs would be to show median values and respective interquartile range (or eventually the range, or other measure to show data dispersion). The use of mean and standard deviation in these cases is not adequate.

Tables 3 and 4 – only the statistically significant coefficients are shown. I suggest including all results in the table, keeping the asterisks indicating when the significance was achieved. Alternatively, at least include hyphens in the empty cells; however, in this option to correct the table title to “Statistically significant coefficients of linear correlation between gene expression in…”

Figure 2 – if possible correct “componente” to “component”

Line 193 – Since the gene expression allows the assumption the tumors were benign neoplasia this results reinforces the suggestion to treat the topic as canine mammary neoplasia (CMNs) and not canine mammary tumors (CMTs) as suggested in this reviewer report in the introduction.

Line 356 – the fact the principal component analysis may have a potential in the diagnosis of malignancies is correct, however, sensitivity and specificity were not analyzed in this study. In this way, the assumption showed in the conclusion (line 481) that the use of this combination of gene analysis can reach up to 80% specificity can not be supported. 

Author Response

This is a nice and well-written study on canine mammary neoplasm gene expression. Some suggestions to improve it are presented below.

Line 15 – the word “Tumors” refers to increased volume despite often being related as neoplasia, technically it can be another pathology process (abscess, cysts, granuloma etc..). In the other hand, “cancer” refers to a malignant neoplasia. In this way, I believe the authors should review the sentence “Canine mammary tumors (CMTs) are the most frequent cancers in intact bitches”

Answer: The changes have been made according to your suggestion

Line 38 – Please modify canine mammary tumors by canine mammary neoplasia.

Answer: It has been changed according to your advice

Lines 50-66 – This paragraph seems to be a mix of discussion about the importance of adopting adequate PCR standards and techniques plus some material and methods when describing the genes used to evaluate normalization. I´m particularly not used to finding this information in the intro.

Answer: Thanks for your comment. This paragraph has been modified.

Line 462 – Shapiro-Wilk results were assumed as non-parametric if the p-value was < 0.05 or below 0.005?

Answer: Thanks for detecting this typo error. It has been corrected in the manuscript.

Line 76 – Despite the material and methods section giving an adequate amount of information regarding molecular biology techniques applied, I believe more detail should be provided in the MMs to inform how the authors assumed the histological classification shown in the table 1. Was any immunohistochemistry applied or the diagnosis was assumed only on HE histologic evaluation?

Answer: Neoplasia classification was performed through HE histologic evaluation. This information has been added to the Material and methods chapter.

Figure 1 – if the data regarding gene expression was assumed as non-parametric, the right way to express them in graphs would be to show median values and respective interquartile range (or eventually the range, or other measure to show data dispersion). The use of mean and standard deviation in these cases is not adequate.

Answer: Figure 1 has been modified according to your suggestion. The data are now expressed as median values and interquartile range. The text has been also modified accordingly.

Tables 3 and 4 – only the statistically significant coefficients are shown. I suggest including all results in the table, keeping the asterisks indicating when the significance was achieved. Alternatively, at least include hyphens in the empty cells; however, in this option to correct the table title to “Statistically significant coefficients of linear correlation between gene expression in…”

Answer: Thanks for the comment. After evaluating the two possibilities, the authors have decided to add hyphens in the empty spaces to avoid the table from becoming overdone.

Figure 2 – if possible correct “componente” to “component”

Answer: Thanks for detecting the mistake. It has been corrected according to your comment.

Line 193 – Since the gene expression allows the assumption the tumors were benign neoplasia this results reinforces the suggestion to treat the topic as canine mammary neoplasia (CMNs) and not canine mammary tumors (CMTs) as suggested in this reviewer report in the introduction.

Answer: CMTs has been changed by CMNs throughout the text.

 Line 356 – the fact the principal component analysis may have a potential in the diagnosis of malignancies is correct, however, sensitivity and specificity were not analyzed in this study. In this way, the assumption showed in the conclusion (line 481) that the use of this combination of gene analysis can reach up to 80% specificity can not be supported.

Answer: Thanks for your comment. This sentence has been removed from the conclusions chapter.

Reviewer 2 Report

Comments and Suggestions for Authors

The manuscript entitled “Selected gene expression in canine mammary neoplasia” encompasses original findings on the gene expression for eight genes VEGFa, CD20, PGR, HYAL-1, PD-L1, EGF, RLN2 and MMP3 in canine mammary tumors. Gene expression assessment was carried out through RT-qPCR. Overall, the article is well written, objectives of the study have successfully been achieved by the authors. Authors are advised to follow a uniform numerical citation style. In lines 388 to 389, Author: Year format has been used instead of numerical format. Necessary corrections shall be made.

Author Response

The manuscript entitled “Selected gene expression in canine mammary neoplasia” encompasses original findings on the gene expression for eight genes VEGFa, CD20, PGR, HYAL-1, PD-L1, EGF, RLN2 and MMP3 in canine mammary tumors. Gene expression assessment was carried out through RT-qPCR. Overall, the article is well written, objectives of the study have successfully been achieved by the authors. Authors are advised to follow a uniform numerical citation style. In lines 388 to 389, Author: Year format has been used instead of numerical format. Necessary corrections shall be made.

Answer: Dear reviewer, thank you very much for your appreciation of our manuscript. The authors also thank you for your observation. The mistake has been corrected.

Reviewer 3 Report

Comments and Suggestions for Authors

The manuscript describes a study to evaluate the expression of several genes selected from animal mammary tumor signaling pathways to identify prognostic or diagnostic cancer markers. The theme is original, The work is well presented and clear, and although the results were different from what was expected, the discussion adequately justifies the results achieved. No combination of parameters provided 100% specificity for differentiating between malignant and benign MTCs. However, the combination of PGR, CD20, EGF, relaxin, and MMP3 provides a slightly above 80% specificity. The work has merit and deserves to be published.

the information shown may be important for several researchers in the field. The discussion adequately justifies the results, and the conclusions are consistent with the evidence and arguments addressing the main question.

The only observation suggests that Table 6 is unnecessary in the text and can be presented as a Supplementary.

Comments on the Quality of English Language

Small grammatical corrections are necessary, but they can be made during editing.

Author Response

The manuscript describes a study to evaluate the expression of several genes selected from animal mammary tumor signaling pathways to identify prognostic or diagnostic cancer markers. The theme is original, The work is well presented and clear, and although the results were different from what was expected, the discussion adequately justifies the results achieved. No combination of parameters provided 100% specificity for differentiating between malignant and benign MTCs. However, the combination of PGR, CD20, EGF, relaxin, and MMP3 provides a slightly above 80% specificity. The work has merit and deserves to be published.

the information shown may be important for several researchers in the field. The discussion adequately justifies the results, and the conclusions are consistent with the evidence and arguments addressing the main question.

The only observation suggests that Table 6 is unnecessary in the text and can be presented as a Supplementary.

Answer: The table has been added as an appendix information following your advice.

Reviewer 4 Report

Comments and Suggestions for Authors

1. The sample size, if possible, should be enlarged.

2. Add more groups, such as cases with tumor but not MT, and case with systemic disease but without tumor.

Author Response

  1. The sample size, if possible, should be enlarged.

Answer: The authors are aware of the limited number of samples, especially in the control group. However, increasing the number of samples hasn’t been possible. Removing a sample of mammary tissue from healthy bitches is not acceptable for many owners despite the permission of the ethical committee. Regarding neoplastic samples, getting a higher number of samples is not easy either due to the low number of CMTs cases we have in our geographical area.

  1. Add more groups, such as cases with tumor but not MT, and case with systemic disease but without tumor.

Answer: The authors appreciate your suggestion. However, adding more subgroups would lead to groups with a very low number of samples. On the other hand, the authors tried to make a follow-up of the patients but, unfortunately, only a few patients were able to be followed due to owners' decision.

Reviewer 5 Report

Comments and Suggestions for Authors

Interesting job. Designed properly. Results presented reading (note below), well discussed.
I only have a few minor comments on the technical side
- figure 2 is illegible. It's supposed to be #D but it's flat. Maybe adding lines, whiskers, to individual axes will improve the situation. There aren't many points, so maybe this will be it... You can also add colors.
- conclusions are a summary of results, not conclusions. Please propose a list of conclusions resulting from the results obtained.

Author Response

Interesting job. Designed properly. Results presented reading (note below), well discussed.

I only have a few minor comments on the technical side

- figure 2 is illegible. It's supposed to be #D but it's flat. Maybe adding lines, whiskers, to individual axes will improve the situation. There aren't many points, so maybe this will be it... You can also add colors.

Answer: Figure 2 has been modified following your suggestions. We hope it’s clearer and easier to understand in its new form.

- conclusions are a summary of results, not conclusions. Please propose a list of conclusions resulting from the results obtained.

Answer: The conclusions have been rewritten following your advice.

Reviewer 6 Report

Comments and Suggestions for Authors

The manuscript is good but requires a few minor revisions. After these, I will support the publication of this manuscript in IJMS. Below are my comments:

  1. Please gently rephrase the title.
  2. The first abstract sentence, which poorly connects with the rest - please review.
  3. The last two sentences of the abstract - the same issue. Please revisit the aim of the study.
  4. "; epidermal growth factor" - is this comment crucial, or does it only appear after improving how growth can be achieved quickly?
  5. The last paragraph of the introduction needs to be rewritten; extract the key objectives of the study and describe the methods used. It cannot remain as it is.
  6. Fig. 2 - needs improvement - very accessible quality and aesthetics.
  7. The discussion is well conducted.
  8. Conclusion - perform a rewrite. Please focus on what the aim of the study was.

The manuscript is very good, and with slight corrections, it will be suitable for publication.

Author Response

The manuscript is good but requires a few minor revisions. After these, I will support the publication of this manuscript in IJMS. Below are my comments:

Please gently rephrase the title.

Answer: The title has been modified following your advice

The first abstract sentence, which poorly connects with the rest - please review.

Answer: Thanks for the comment. The sentence indeed was poorly connected. It has been removed and the following sentence has been modified accordingly.

The last two sentences of the abstract - the same issue. Please revisit the aim of the study.

Answer: The abstract has been modified.

"; epidermal growth factor" - is this comment crucial, or does it only appear after improving how growth can be achieved quickly?

Answer: The authors are not completely sure of this question. Epidermal growth factor has been added to keywords, as PD-L1, relaxin and MMP3 have been, because of the significant results obtained in the study.

The last paragraph of the introduction needs to be rewritten; extract the key objectives of the study and describe the methods used. It cannot remain as it is.

Answer: The last paragraph of the introduction has been slightly modified. We hope it is acceptable now.

Fig. 2 - needs improvement - very accessible quality and aesthetics.

Answer: Figure 2 has been modified following your suggestions. We hope it’s clearer and easier to understand in its new form.

The discussion is well conducted.

Answer: thank you very much for your appreciation.

Conclusion - perform a rewrite. Please focus on what the aim of the study was.

The manuscript is very good, and with slight corrections, it will be suitable for publication.

Answer: The conclusions have been rewritten according to your suggestion.

Round 2

Reviewer 5 Report

Comments and Suggestions for Authors

the authors have made corrections and now the manuscript can be considered for publication

Reviewer 6 Report

Comments and Suggestions for Authors

The manuscript is highly acceptable. The authors have revised it appropriately. I believe it can be accepted.